# Trajectories of disability and influence of contextual factors among adults aging with HIV: Insights from a community-based longitudinal study in Toronto, Canada

Tai-Te Su[1,2], Ahmed M. Bayoumi[3,4,5,6], Lisa Avery[7], Soo Chan Carusone[1,8], Ada Tang[8,9], Patricia Solomon[9], Aileen M. Davis[1,6,10], Kelly K. O'Brien[1,6,10]*

1 Department of Physical Therapy, University of Toronto, Toronto, Ontario, Canada, 2 School and Graduate Institute of Physical Therapy, College of Medicine, National Taiwan University, Taipei, Taiwan, 3 MAP Centre for Urban Health Solutions, Li Ka Shing Knowledge Institute, St. Michael's Hospital, Toronto, Ontario, Canada, 4 Division of General Internal Medicine, St. Michael's Hospital, Toronto, Ontario, Canada, 5 Department of Medicine, Temerty Faculty of Medicine, University of Toronto, Toronto, Ontario, Canada, 6 Institute of Health Policy, Management and Evaluation (IHPME), University of Toronto, Toronto, Ontario, Canada, 7 University Health Network, Toronto, Ontario, Canada, 8 McMaster Collaborative for Health and Aging, McMaster University, Hamilton, Ontario, Canada, 9 School of Rehabilitation Science, Faculty of Health Sciences, McMaster University, Hamilton, Ontario, Canada, 10 Rehabilitation Sciences Institute (RSI), University of Toronto, Toronto, Ontario, Canada

* kelly.obrien@utoronto.ca

## Abstract

### Background

Individuals aging with HIV may experience disability that is multidimensional and evolving over time. Our aims were to characterize the longitudinal trajectories of disability and to investigate how intrinsic and extrinsic contextual factors influence dimensions of disability over an eight-month period among adults aging with HIV.

### Methods

We analyzed longitudinal observational data from a community-based study in Toronto, Canada, where adults aging with HIV completed self-reported questionnaires over eight months (five time points). We measured disability using the Short-Form HIV Disability Questionnaire (SF-HDQ), which included six dimensions: physical, cognitive, mental-emotional health challenges, uncertainty, difficulties with day-to-day activities, and challenges to social inclusion. Higher SF-HDQ scores (range: 0–100) indicate greater severity of disability. We assessed intrinsic (age, gender, education, living status, number of comorbidities, mastery) and extrinsic (stigma, social support) contextual factors using baseline self-reported questionnaires. Latent class growth analysis was performed to identify distinct disability trajectories within each of the six dimensions. Multinomial logistic regression models were used to assess the influence of contextual factors on the disability trajectories.

**Data availability statement:** Data cannot be shared publicly as participants did not consent to providing public access to their data in the consent process. All relevant data supporting the findings are within the manuscript and its Supporting Information files. For more information regarding data availability for this study, individuals may contact the Research Ethics Board, University of Toronto (ethics.review@utoronto.ca).

**Funding:** This study was funded by the Canadian Institutes of Health Research (CIHR) HIV/AIDS Community-Based Research (CBR) Program (Funding Reference Number #CBR-139685; 160 Elgin Street, Ottawa, Ontario, Canada, K1A 0W9). https://cihr-irsc.gc.ca/e/193.html. Tai-Te Su was supported by the Ontario HIV Treatment Network Endgame Research Program – Breaking New Ground Award (EFP-1121-BNG) (https://www.ohtn.on.ca/). Kelly K. O'Brien is supported by a Canada Research Chair in Episodic Disability and Rehabilitation from the Canada Research Chairs Program (CRC-2002-00510) (https://www.chairs-chaires.gc.ca/home-accueil-eng.aspx). Ada Tang was supported by a Clinician-Scientist Award (Phase II) from the Ontario Heart & Stroke Foundation (P-19-TA-1192). Ahmed M. Bayoumi was supported by the Fondation Alma and Baxter Ricard Chair in Inner City Health at St. Michael's Hospital and the University of Toronto. The funders had no role in study design, data collection and analysis, decision to publish, or preparation of the manuscript.

**Competing interests:** The authors have declared that no competing interests exist.

## Results

Of 108 participants, 89% identified as men with a mean age of 50.6 years (standard deviation ±10.9). We identified three disability trajectories: low, medium, and high disability severity in the physical, mental-emotional, and day-to-day activities dimensions. Four trajectories: low, medium-low, medium-high, and high disability severity were in the cognitive, uncertainty, and social inclusion dimensions. Factors such as higher self-mastery and social support were associated with lower disability trajectories, whereas greater number of comorbidities and stigma were associated with more severe disability trajectories over time.

## Conclusion

Disability experiences among adults aging with HIV included three or four distinct trajectories with considerable heterogeneity over time. Information on contextual factors may be helpful for informing interventions and supports that mitigate disability among adults aging with HIV.

## Introduction

With advancements in medical treatments and healthcare services, people living with human immunodeficiency virus (HIV) are living longer [1–3]. Population-based studies in Europe and North America have shown that the average life expectancy of people living with HIV who have access to care is nearing that of the general population [3,4]. While HIV is now considered a chronic illness, people aging with HIV may experience a wide range of challenges such as adverse treatment effects (e.g., from prolonged antiretroviral treatment), comorbidities (e.g., cardiovascular diseases, metabolic syndrome, neurocognitive disorders, or cancer), insecure employment or unemployment, and HIV-associated stigma and lack of social support [5–12]. Informed by the experiences of individuals living with HIV, the *Episodic Disability Framework* characterizes these health-related challenges as disability, and provides a conceptual foundation to illustrate its multidimensional and episodic nature [13]. The Framework includes six dimensions: physical, cognitive, mental or emotional symptoms, uncertainty or worry about the future, difficulties with day-to-day activities, and challenges to social inclusion [13,14]. The Framework also asserts that the severity and presence of disability are sometimes experienced as episodic in nature, which can be exacerbated or alleviated by a range of intrinsic contextual factors (i.e., personal attributes and living strategies) and extrinsic contextual factors (i.e., stigma and social support) [14].

To develop effective interventions and services, it is critical to understand the evolving nature of disability and its influencing factors experienced by people aging with HIV. Studies have attempted to delineate the patterns of change in disability among this population [15–21]. For instance, Crystal and Sambamoorthi identified an increase in the number of basic daily activities that individuals living with HIV

had difficulty performing independently (0.32 additional tasks per month) over a two-year follow-up period [15]. Meanwhile, Brouillette and colleagues found stable trajectories of neurocognitive function among people living with HIV, but noted heterogeneity over three years [19]. Furthermore, a one-year prospective study conducted in an outpatient clinic setting revealed a gradual decrease in positive affect (e.g., mood) among people living with HIV, whereas negative affect remained unchanged [21]. While this evidence provides critical insights, each of these studies focused on a single dimension of disability, such as difficulties with day-to-day activities or mental or emotional symptoms, as outlined in the *Episodic Disability Framework*. To our knowledge, no published study has examined the longitudinal profiles of disability using a multidimensional approach. As the impact of aging with HIV spans various life domains [22], it is important to adopt a holistic approach to understand the needs and challenges experienced by people aging with this chronic illness.

Guided by the *Episodic Disability Framework*, the primary objective of this study was to characterize longitudinal trajectories of disability among adults aging with HIV across six different dimensions. The secondary objective was to assess the influence of intrinsic contextual factors (age, gender, education, living status, number of comorbidities, mastery) and extrinsic contextual factors (stigma, social support), on disability trajectories over time. Findings will contribute to an enhanced understanding of disability among people aging with HIV, and will provide empirical evidence to advance care, programs, and services aimed at reducing disability and improving health outcomes among this population.

## Materials and methods

### Study design and participants

We conducted a longitudinal observational study using data from a community-based exercise (CBE) intervention study involving adults aging with HIV in Toronto, Canada. The CBE intervention study was a three-phased study spanning 22 months, originally designed to assess the effectiveness of a CBE intervention on cardiopulmonary fitness, strength, flexibility, and physical activity among adults aging with HIV [23]. Adults 18 years or older with HIV, who were considered medically stable and safe to participate in exercise, were recruited from HIV community-based organizations and the Toronto YMCA between July 29, 2016 and January 26, 2017. For the first eight months, participants were monitored every two months without any interventions. After this baseline monitoring phase, participants engaged in thrice-weekly supervised exercise sessions followed by self-monitored exercise. In this study, we examined disability trajectories during the eight-month baseline (pre-intervention) monitoring phase.

### Ethics and consent

The study was registered (registration number: NCT02794415) and received approval from the Research Ethics Board at the University of Toronto (Protocol #32910). All participants provided written informed consent to participate in the study. Detailed information on study design and trial protocol has been published [23].

### Data sources

We used bimonthly observational data collected during the eight-month baseline monitoring phase, whereby participants completed a series of self-reported electronic administered questionnaires at baseline, 2 months, 4 months, 6 months and 8 months (five time points), to characterize experiences of disability over time prior to exercise intervention. See S1 Fig. for an overview of the data sources and measurement timepoints.

### Measures

**Disability.** Disability was assessed at each time point using the electronic version of the Short-Form HIV Disability Questionnaire (SF-HDQ) [24]. Derived from the original 69-item HDQ [25,26], the SF-HDQ is a patient-reported outcome measure containing 35 items that assess disability across six different dimensions: physical symptoms (ten

items), cognitive symptoms (three items), mental and emotional symptoms (five items), uncertainty about future health (five items), difficulties carrying out day-to-day activities (five items), and challenges to social inclusion (seven items). Scores for each dimension are derived from a Rasch-based logit scale ranging from 0 to 100 whereby higher scores indicate greater severity of disability [24]. A single summary score for disability is not provided by the SF-HDQ due to the unidimensionality assumption of the Rasch model and the distinct nature of each disability dimension. At the end of the questionnaire, participants indicated whether they perceive they completed the SF-HDQ on a "good day" or "bad day" living with HIV. The SF-HDQ is an internally consistent instrument (Cronbach's α = 0.78–0.85) and possesses sensibility and utility for use in clinical and community-based settings among adults living with HIV [27,28].

### Intrinsic contextual factors

The selection of contextual factors was guided by the *Episodic Disability Framework* [13]. We included self-mastery and personal attributes measured at baseline as intrinsic contextual factors in this study. Levels of self-mastery were assessed using the Pearlin Mastery Scale [29]. The scale contains a total of seven items, including two positively worded state-ments (e.g., *What happens to me in the future mostly depends on me*) and five negatively worded statements (e.g., *There is really no way I can solve some of the problems I have*). Participants self-reported their agreement with each statement on a four-point Likert scale (strongly agree [1], agree [2], disagree [3], and strongly disagree [4]). Positively worded ques-tions were reverse coded. Final scores were summed and ranged from 7 to 28, with higher scores representing greater self-mastery. The Pearlin Mastery Scale has demonstrated reliability and construct validity for caregivers of older adults and persons living with HIV [30,31].

Information on personal attributes, including age (years), gender (man vs. woman; only these categories were repre-sented in the study), education (some university education or higher vs. otherwise), living status (living alone vs. living with others), and number of comorbidities was gathered through an electronically administered demographic questionnaire at baseline.

### Extrinsic contextual factors

Extrinsic contextual factors included self-perceived stigma and social support measured at baseline. We measured stigma using the 40-item HIV Stigma Scale, a self-administered questionnaire designed to assess HIV-related stigma across four subscales, including personalized stigma, disclosure concerns, negative self-image, and concern with public attitudes [32]. Participants self-reported their agreement with each item on a four-point Likert scale (strongly disagree [1], disagree [2], agree [3], and strongly agree [4]). Final scores ranged from 40 to 160, with higher scores indicating greater HIV stigma. The HIV Stigma Scale demonstrates reliability (test-retest reliability α = 0.96) and construct validity with people living with HIV [32,33].

We measured perceived social support using the 19-item Medical Outcomes Study Social Support Survey (MOS-SSS), a self-administered questionnaire designed for community-dwelling individuals with chronic conditions [34]. The ques-tionnaire contains four domains: emotional/information support, tangible support, affectionate support, and positive social interaction. Participants reported the frequency of receiving each type of support on a five-point Likert scale, ranging from none of the time (0) to all of the time (5). Following the established procedure, raw scores across the four domains were summed and transformed into a 100-point scale where higher scores represent greater social support [34]. The MOS-SSS has demonstrated internal-consistency reliability and construct validity among people living with HIV [35,36].

### Analysis

We described participants' personal and clinical characteristics at baseline. Mean and standard deviation were reported for continuous variables, while number and percentage were reported for categorical variables.

We conducted latent class growth analysis (LCGA), also known as group-based trajectory modeling, to identify trajectories across six disability dimensions as measured by the SF-HDQ over the eight-month baseline monitoring period [37]. LCGA is a specialized application of the semi-parametric finite mixture modeling where the population is assumed to consist of a finite number of unobserved latent groups, with individuals in each group exhibiting similar pattern or progression of outcomes over time [37,38]. Unlike basic growth curve analysis where a single trajectory was estimated by pooling individuals across the entire population, LCGA represents a flexible statistical procedure that is able to better characterize the heterogeneity in health outcomes, including disability over time. In particular, this approach allows researchers to identify specific subgroups with unique health-related challenges (e.g., dimensions of disability) or profiles and has the potential to inform personalized intervention programs and services [39]. To account for variability in the timing of questionnaire completion caused by scheduling issues [40], we adjusted the time variable to accurately reflect the actual number of days between assessments. A quadratic link function for time was specified for the trajectory models. We determined the optimal number of trajectories for each disability dimension based on the parsimony principal and a combination of the following criteria: 1) lowest value of Bayesian Information Criteria (BIC), 2) group size not less than 5% of the total sample, and 3) the average posterior probability of group assignment (≥0.7) [37]. Missing data were handled under the assumption of missing at random and accommodated using maximum likelihood estimations [41]. As a final step, the labeling of each disability trajectory was completed through joint consideration of model parameters (e.g., intercept and slopes) and relative comparisons with other trajectories identified within the same dimension.

After participants were assigned to their respective trajectory group in each of the six disability dimensions, we conducted multinomial logistic regression models to assess the influence of intrinsic contextual factors (age, gender, education, living status, number of comorbidities, mastery) and extrinsic contextual factors (stigma, social support) measured at baseline on subsequent disability trajectories. A total of six regression models were tested, with each focusing on a specific disability dimension. Considering the potential difficulty in interpreting relative comparisons such as relative risk ratio (RRR) to a specific trajectory group, we calculated marginal effects from the regression models to illustrate the associations between contextual factors and the predicted probabilities of each disability trajectory. A two-tailed $p$-value $< 0.05$ was considered statistically significant. All analyses were performed in the R software (Version 4.3.1).

## Results

### Participant characteristics at baseline

Out of the 120 recruited adults, 108 consented to participate and initiated the baseline monitoring phase. Out of these 108 participants, 83 (77%) remained in the study at the end of the baseline monitoring phase (month 8). The present study comprised a total of 436 person-month of observations, with each participant completing an average of 4.0 (standard deviation [SD] ±1.4) assessments out of 5.

Table 1 includes participant characteristics and contextual factors among the 108 participants at baseline. Overall, the average age was 50.6±10.9 years, and the majority were identified as White (63%), men (89%), and living alone (68%). The average duration since HIV diagnosis was 17.5±9.9 years, with nearly all participants (99%) taking antiretroviral medications. Approximately half of the participants received some university degree or higher (45%), and they had an average of five comorbidities in addition to living with HIV. The three most prevalent comorbidities were mental health conditions (n=52), joint pain (n=44), and bone and joint disorders (n=39). The average scores for self-mastery, total HIV-related stigma, and total social support were 19.7±4.0 (out of 28), 93.6±24.6 (out of 160), and 53.3±24.8 (out of 100), respectively. SF-HDQ disability severity domain scores ranged from 15.1±13.6 (difficulties with day-to-day activities) to 38.4±22.5 (uncertainty) on a 100-point scale at baseline (Table 2). Most participants (82%) reported having a 'good day' when they completed the SF-HDQ. Characteristics of participants who remained in the study at the end of the baseline monitoring phase (n=83) is detailed in S1 Table.

**Table 1. Characteristics of adults living with HIV and contextual factors at baseline (n = 108).**

| Participant characteristics | Mean ± SD/ n (%) |
|---|---|
| Age (years) | 50.6 ± 10.9 |
| Gender (n) | |
| Men | 96 (89) |
| Women | 12 (11) |
| Race/Ethnicity (n) | |
| White | 68 (63) |
| Black or African | 9 (8) |
| Hispanic or Latino | 5 (5) |
| Number of years since HIV diagnosis | 17.5 ± 9.9 |
| Self-reported undetectable HIV viral load (<50 copies/mL) | 90 (83) |
| Current use of antiretroviral medications (yes; n) | 107 (99) |
| Education (some university+) | 49 (45) |
| Living alone (n) | 73 (68) |
| Number of comorbidities in addition to living with HIV (count) | 4.9 ± 4.1 |
| Three most common self-reported comorbidities | |
| Mental health condition (e.g., depression, anxiety) | 52 (48) |
| Joint pain (arthritis) | 44 (41) |
| Bone and joint disorder (e.g., osteonecrosis, osteopenia) | 39 (36) |
| Pearlin Mastery Scale score (range: 7–28) | 19.7 ± 4.0 |
| HIV Stigma Scale total score (range: 40–160) † | 93.6 ± 24.6 |
| Personalized stigma subscale score (range: 18–72) | 40.7 ± 12.0 |
| Disclosure concerns subscale score (range: 10–40) | 25.3 ± 7.2 |
| Negative self-image subscale score (range: 13–52) | 27.6 ± 8.6 |
| Public attitudes subscale score (range: 20–80) | 47.0 ± 13.0 |
| MOS Social Support Survey total score (range: 0–100) ‡ | 53.3 ± 24.8 |
| Emotional and information support raw score (range: 8–40) | 26.1 ± 7.8 |
| Tangible support raw score (range: 4–20) | 11.7 ± 5.7 |
| Affectionate support raw score (range: 3–15) | 9.1 ± 4.1 |
| Positive social interaction raw score (range: 4–20) | 12.7 ± 4.6 |

SD = Standard deviation.

†Sixteen items on the HIV Stigma Scale belong to more than one subscale, which reflects the intercorrelations between different subscales.

‡ Raw scores of the MOS Social Support Survey were summed across the four subdomains and transformed into a 100-point scale. Greater Pearlin Mastery Scale scores, HIV Stigma Scale scores, and MOS Social Support Survey scores reflect higher levels of self-mastery, HIV-related stigma, and perceived social support, respectively.

## Disability trajectories across six dimensions over eight months

Fig 1 portrays the trajectories of disability across six dimensions, highlighting the substantial variability within and between participants' experiences over time. Results from the LCGA suggested that a model with three trajectory groups provided the best fit for the dimensions of physical symptoms, mental-emotional symptoms, and difficulties with day-to-day activities, whereas a model with four trajectory groups exhibited the best bit for the dimensions of cognitive symptoms, uncertainty, and challenges to social inclusion (Table 3). The three-group trajectories were labeled as low, medium and high disability severity, and the four-group trajectories were labeled as low, medium-low, medium-high, and high disability severity over time.

**Table 2. Short-Form HIV Disability Questionnaire (SF-HDQ) scores at baseline (n = 108).**

| SF-HDQ domains | Mean ± SD/ n (%) |
|---|---|
| Physical symptoms severity scores | 30.2 ± 13.8 |
| Cognitive symptoms severity scores | 23.5 ± 20.2 |
| Mental-emotional symptoms severity scores | 35.6 ± 20.8 |
| Uncertainty severity scores | 38.4 ± 22.5 |
| Difficulties with day-to-day activities severity scores | 15.1 ± 13.6 |
| Challenges to social inclusion severity scores | 33.1 ± 14.8 |
| Good day-bad day item | n (%) |
| Good day (%) | 89 (82) |
| Bad day (%) | 19 (18) |

SD = Standard deviation. Severity scores were on a Rasch-based logit scale ranging from 0 to 100, where higher scores indicate greater disability severity. Participants self-reported whether they had a good day or bad day living with HIV when they completed the SF-HDQ.

For each disability dimension, participants were assigned to one specific trajectory group based on the highest posterior probability and thus their group membership remained unchanged over the eight-month timeframe. For physical symptoms, 37% of participants showed a low disability trajectory (mean SF-HDQ score:12.1 ± 10.0), while 34% and 29% showed medium (26.0 ± 13.0) and high (32.0 ± 19.8) disability trajectories over time. For mental-emotional symptoms, 38% of participants exhibited a low (14.0 ± 12.4) disability trajectory, while 32% and 30% exhibited medium (30.0 ± 16.7) and high (39.4 ± 27.6) disability trajectories in this dimension. For difficulties with day-to-day activities, 36% showed a low (2.6 ± 5.5) disability trajectory, whereas 46% and 18% showed medium (14.3 ± 10.4) and high (25.8 ± 17.0) disability trajectories, respectively. For cognitive symptoms, 30% of participants demonstrated a low disability trajectory (3.9 ± 6.9), and the rest displayed medium-low (14.0 ± 11.4; 33%), medium-high (28.4 ± 16.7; 31%), and high (32.6 ± 34.3; 7%) trajectories. For the uncertainty dimension, 15% of participants exhibited a low (5.7 ± 9.5) disability trajectory, while the majority showed medium-low (25.5 ± 15.3; 51%), medium-high (40.2 ± 21.8; 26%), and high (53.5 ± 34.9; 8%) trajectories. For challenges to social inclusion, participants demonstrated low (8.1 ± 9.0; 24%), medium-low (22.1 ± 11.6; 21%), medium-high (30.3 ± 17.4; 42%), and high (43.5 ± 18.6; 13%) trajectories over time. The interconnections between trajectories across the six disability dimensions were visually illustrated in S2 Fig. Detailed information on each disability trajectory, including intercepts and slopes, is provided in S2 Table and S3 Table. The average posterior probability of group assignment ranged from 0.86 to 0.98 across all trajectory groups, surpassing the standard of ≥0.7 (S4 Table). Throughout the eight-month monitoring phase, 60.2% of participants consistently reported having 'good days' when they completed the SF-HDQ, while 1.9% consistently reported 'bad days' across all bimonthly assessments. Approximately 38% reported experiencing both good and bad days at different time points during this period.

## Influences of contextual factors on disability dimension trajectories

Table 4 and Table 5 summarize the influences of intrinsic and extrinsic contextual factors on the six dimensions of disability.

Overall, a decade increase in age was associated with a 12% increase in the probability of being allocated to the medium disability severity trajectory for physical symptoms (*pr*[Medium]: 0.12 [0.00, 0.24]). Compared with women, men were 28% and 7% more likely to be allocated to the high disability severity trajectory for physical symptoms (*pr*[High]: 0.28 [0.20, 0.37]) and cognitive symptoms (*pr*[High]: 0.07 [0.01, 0.12]), respectively. In addition, compared with women, men were more likely to be allocated to the medium-low trajectory for challenges to social inclusion (*pr*[Medium-Low]: 0.18 [0.10, 0.27]) and the medium trajectory for mental-emotional symptoms (*pr*[Medium]: 0.33 [0.22, 0.44]). Participants with some university degree or higher were less likely to be allocated to the high severity

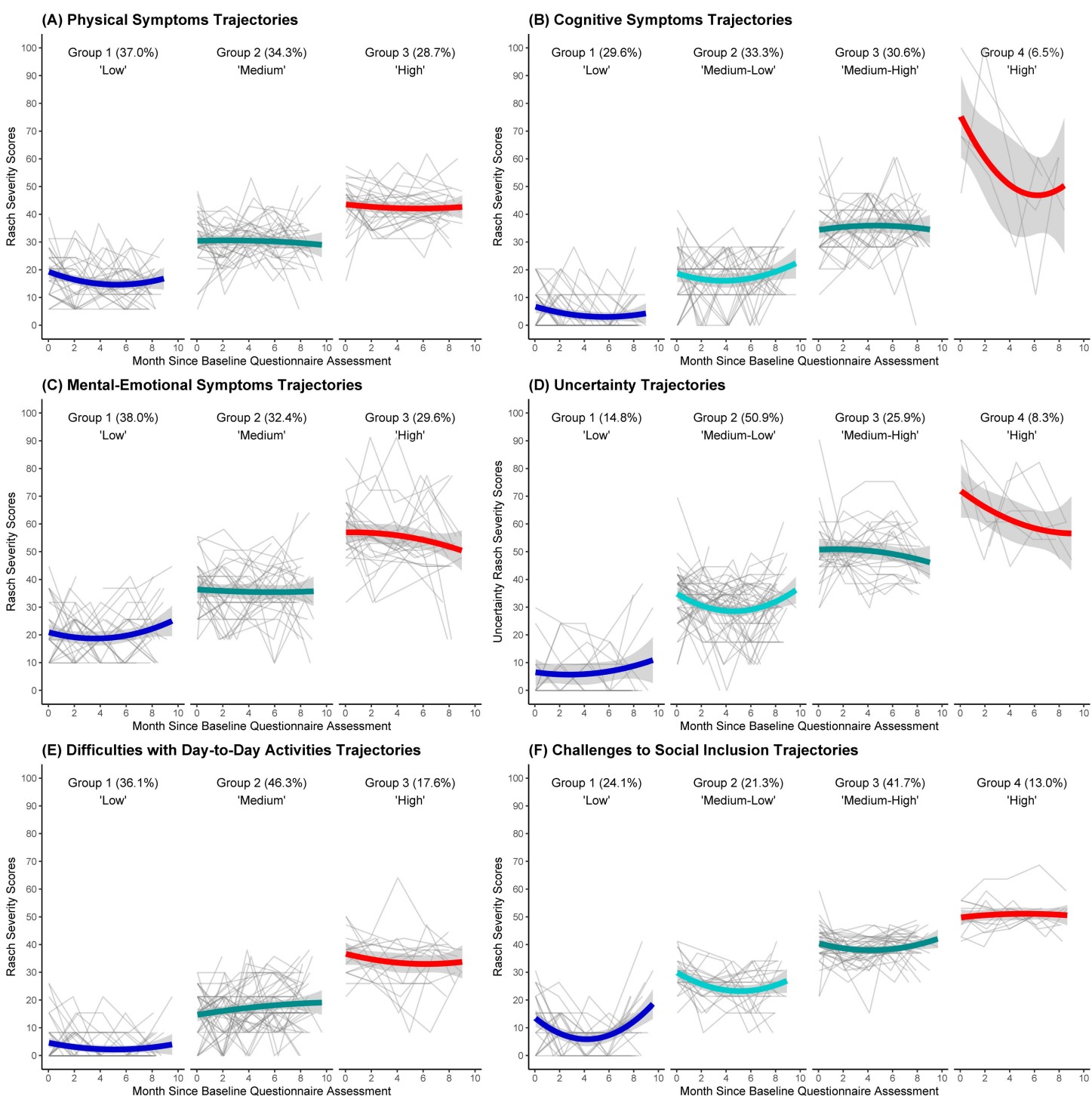

**Fig 1. Trajectories of disability across six dimensions among adults living with HIV over eight months.** Each bolded line corresponds to a unique trajectory group identified within each of the six disability dimensions. Each color represents a distinct disability trajectory group. The 95% confidence intervals are depicted as shaded areas.

**Table 3. Model fit statistics for disability trajectories across six dimensions among adults living with HIV over eight months.**

| Disability dimensions | Number of Trajectories | BIC | Posterior classification of trajectory memberships n (%) | | | | |
|---|---|---|---|---|---|---|---|
| | | | Group 1 | Group 2 | Group 3 | Group 4 | Group 5 |
| Physical symptoms | 1 | 3543.0 | 108 (100) | | | | |
| | 2 | 3378.9 | 62 (57.4) | 46 (42.6) | | | |
| | **3** | **3357.5** | **37 (34.3)** | **40 (37.0)** | **31 (28.7)** | | |
| | 4 | 3363.5 | 33 (30.6) | 31 (28.7) | 4 (3.7) | 40 (37.0) | |
| Cognitive symptoms | 1 | 3766.3 | 108 (100) | | | | |
| | 2 | 3589.1 | 41 (38.0) | 67 (62.0) | | | |
| | 3 | 3540.3 | 8 (7.4) | 38 (35.2) | 62 (57.4) | | |
| | **4** | **3528.4** | **7 (6.5)** | **36 (33.3)** | **33 (30.6)** | **32 (29.6)** | |
| | 5 | 3544.5 | 7 (6.5) | 30 (27.8) | 32 (29.6) | 7 (6.5) | 32 (29.6) |
| Mental-emotional symptoms | 1 | 3858.1 | 108 (100) | | | | |
| | 2 | 3682.4 | 63 (58.3) | 45 (41.7) | | | |
| | **3** | **3663.3** | **41 (38.0)** | **35 (32.4)** | **32 (29.6)** | | |
| | 4 | 3646.4 | 38 (35.2) | 34 (31.5) | 31 (28.7) | 5 (4.6) | |
| Uncertainty | 1 | 3868.9 | 108 (100) | | | | |
| | 2 | 3738.4 | 40 (37.0) | 68 (63.0) | | | |
| | 3 | 3644.0 | 63 (58.3) | 17 (15.7) | 28 (25.9) | | |
| | **4** | **3618.2** | **16 (14.8)** | **9 (8.3)** | **28 (25.9)** | **55 (50.9)** | |
| | 5 | 3609.3 | 54 (50.0) | 16 (14.8) | 2 (1.9) | 11 (10.2) | 25 (23.1) |
| Difficulties with day-to-day activities | 1 | 3513.5 | 108 (100) | | | | |
| | 2 | 3362.1 | 32 (29.6) | 76 (70.4) | | | |
| | **3** | **3305.6** | **50 (46.3)** | **39 (36.1)** | **19 (17.6)** | | |
| | 4 | 3307.5 | 39 (36.1) | 13 (12.0) | 49 (45.4) | 7 (6.5) | |
| Challenges to social inclusion | 1 | 3644.4 | 108 (100) | | | | |
| | 2 | 3353.2 | 73 (67.6) | 35 (32.4) | | | |
| | 3 | 3268.2 | 26 (24.1) | 25 (23.2) | 57 (52.8) | | |
| | **4** | **3233.8** | **14 (13.0)** | **23 (21.3)** | **45 (41.7)** | **26 (24.1)** | |
| | 5 | 3235.6 | 8 (7.4) | 45 (41.7) | 14 (13.0) | 20 (18.5) | 21 (19.4) |

BIC = Bayesian Information Criterion. The final model selected for each disability dimension was marked in bold.

trajectory for physical symptoms ($pr$[High]: −0.21 [−0.37, −0.04]), cognitive symptoms ($pr$[High]: −0.10 [−0.19, −0.02]), and uncertainty ($pr$[High]: −0.10 [−0.15, −0.05]) compared with those without such educational attainment. Compared with those living with others, participants who lived alone were more likely to be allocated to the high severity trajectory for cognitive symptoms ($pr$[High]: 0.10 [0.04, 0.15]) but less likely to be allocated to the high trajectory for uncertainty ($pr$[High]: −0.13 [−0.13, −0.12]). Regarding comorbidities, each additional comorbidity was associated with an increased probability of allocation to high or medium-high trajectories for physical ($pr$[High]: 0.06 [0.02, 0.09]) and cognitive symptoms ($pr$[Medium-High]: 0.03 [0.00, 0.06]). Meanwhile, a greater number of comorbidities also was associated with a decreased probability of allocation to the low or medium-low trajectories for uncertainty ($pr$[Low]: −0.03 [−0.05, −0.01]), difficulties with day-to-day activities ($pr$[Low]: −0.05 [−0.08, −0.01]), and challenges to social inclusion ($pr$[Medium-Low]: −0.03 [−0.06, −0.00]). Participants with a higher level of self-mastery were less likely to be allocated to the high or medium-high trajectories for physical symptoms ($pr$[High]: −0.31 [−0.58, −0.04]), cognitive symptoms ($pr$[High]: −0.31 [−0.56, −0.06]), mental-emotional symptoms ($pr$[High]: −0.29 [−0.58, −0.01]), and uncertainty over time ($pr$[Medium-High]: −0.40 [−0.62, −0.17]).

**Table 4. Influence of contextual factors on the predicted probabilities of disability trajectories (disability dimensions with three trajectories).**

| | Low (L) trajectory | | Medium (M) trajectory | | High (H) trajectory | |
|---|---|---|---|---|---|---|
| | **Physical symptoms** | | | | | |
| **Contextual factors** | **Pr(L)** | **p** | **Pr(M)** | **p** | **Pr(H)** | **p** |
| Age | −0.01 [−0.12, 0.10] | .81 | 0.12 [0.00, 0.24] | .04 | −0.11 [−0.23, 0.02] | .09 |
| Gender: Men | −0.27 [−0.61, 0.07] | .12 | −0.01 [−0.37, 0.34] | .93 | 0.28 [0.20, 0.37] | <.001 |
| University or higher | 0.10 [−0.11, 0.31] | .35 | 0.11 [−0.10, 0.31] | .32 | −0.21 [−0.37, −0.04] | .02 |
| Living alone | −0.11 [−0.32, 0.10] | .32 | −0.03 [−0.26, 0.19] | .77 | 0.14 [−0.02, 0.30] | .08 |
| Comorbidities (count) | −0.05 [−0.08, −0.02] | .002 | −0.01 [−0.04, 0.03] | .69 | 0.06 [0.02, 0.09] | <.001 |
| Self-mastery | 0.23 [−0.06, 0.53] | .12 | 0.07 [−0.24, 0.38] | .65 | −0.31 [−0.58, −0.04] | .03 |
| HIV stigma | −0.03 [−0.08, 0.02] | .27 | 0.07 [0.02, 0.12] | .01 | −0.04 [−0.09, 0.01] | .11 |
| Social support | 0.02 [−0.03, 0.06] | .46 | 0.02 [−0.02, 0.06] | .37 | −0.04 [−0.08, 0.00] | .07 |
| | **Mental-emotional symptoms** | | | | | |
| **Contextual factors** | **Pr(L)** | **p** | **Pr(M)** | **p** | **Pr(H)** | **p** |
| Age | 0.07 [−0.03, 0.17] | .17 | −0.04 [−0.15, 0.08] | .54 | −0.03 [−0.14, 0.07] | .53 |
| Gender: Men | −0.45 [−0.68, −0.22] | <.001 | 0.33 [0.22, 0.44] | <.001 | 0.12 [−0.10, 0.35] | .28 |
| University or higher | 0.08 [−0.13, 0.29] | .45 | −0.01 [−0.24, 0.21] | .92 | −0.07 [−0.27, 0.13] | .50 |
| Living alone | −0.08 [−0.29, 0.13] | .46 | −0.05 [−0.26, 0.18] | .67 | 0.13 [−0.05, 0.31] | .16 |
| Comorbidities (count) | −0.01 [−0.04, 0.02] | .63 | 0.01 [−0.02, 0.04] | .45 | −0.00 [−0.03, 0.03] | .76 |
| Self-mastery | 0.34 [0.05, 0.63] | .02 | −0.05 [−0.36, 0.26] | .75 | −0.29 [−0.58, −0.01] | .04 |
| HIV stigma | −0.01 [−0.06, 0.04] | .71 | 0.01 [−0.04, 0.07] | .67 | −0.00 [−0.05, 0.05] | .92 |
| Social support | 0.04 [0.00, 0.08] | .03 | 0.01 [−0.04, 0.05] | .80 | −0.05 [−0.08, −0.02] | .004 |
| | **Difficulties with day-to-day activities** | | | | | |
| **Contextual factors** | **Pr(L)** | **p** | **Pr(M)** | **p** | **Pr(H)** | **p** |
| Age | −0.00 [−0.10, 0.10] | .99 | −0.05 [−0.17, 0.07] | .41 | 0.05 [−0.04, 0.14] | .27 |
| Gender: Men | 0.04 [−0.32, 0.40] | .83 | −0.09 [−0.51, 0.32] | .66 | 0.05 [−0.18, 0.28] | .65 |
| University or higher | 0.17 [−0.05, 0.39] | .13 | −0.05 [−0.29, 0.19] | .68 | −0.12 [−0.28, 0.05] | .16 |
| Living alone | 0.11 [−0.09, 0.32] | .28 | −0.07 [−0.32, 0.17] | .54 | −0.04 [−0.21, 0.13] | .65 |
| Comorbidities (count) | −0.05 [−0.08, −0.01] | .01 | 0.03 [−0.01, 0.06] | .10 | 0.02 [−0.00, 0.04] | .06 |
| Self-mastery | 0.19 [−0.10, 0.49] | .20 | 0.02 [−0.32, 0.35] | .92 | −0.21 [−0.44, 0.02] | .07 |
| HIV stigma | 0.00 [−0.05, 0.05] | .92 | −0.03 [−0.08, 0.03] | .36 | 0.02 [−0.01, 0.06] | .22 |
| Social support | 0.01 [−0.03, 0.06] | .53 | 0.00 [−0.05, 0.05] | .99 | −0.01 [−0.05, 0.02] | .43 |

Marginal mean probabilities were calculated from the multinomial logistic regression models. Pr() = change in the probability of being allocated to a certain trajectory group. The 95% confidence intervals were presented in square brackets. To facilitate interpretation, we reported the results for contextual factors, including age, self-mastery, HIV stigma scores, and social support, scaled by a factor of 10 to interpret the change in predicted probabilities in terms of a decade increase or a 10-point increase in these factors.

For stigma, a higher score on the HIV Stigma Scale (indicating greater stigma) was associated with an increased probability of being allocated to the medium or high trajectories for physical symptoms (*pr*[Medium]: 0.07 [0.02, 0.12]) and challenges to social inclusion (*pr*[High]: 0.04 [0.01, 0.07]). Additionally, higher HIV Stigma scores were associated with a decreased probability of being allocated to the low trajectory for uncertainty (*pr*[Low]: −0.06 [−0.09, −0.03]) and high trajectory for cognitive symptoms (*pr*[High]: −0.04 [−0.07, −0.01]). For social support, participants with greater self-perceived social support were more likely to be allocated to the low disability trajectory for mental-emotional symptoms (*pr*[Low]: 0.04 [0.00, 0.08]), uncertainty (*pr*[Low]: 0.04 [0.01, 0.06]), and challenges to social inclusion (*pr*[Low]: 0.05 [0.01, 0.08]) over time.

**Table 5. Influence of contextual factors on the predicted probabilities of disability trajectories (disability dimensions with four trajectories).**

| Contextual factors | Low (L) trajectory | | Medium-low (M-L) trajectory | | Medium-high (M-H) trajectory | | High (H) trajectory | |
|---|---|---|---|---|---|---|---|---|
| | **Cognitive symptoms** | | | | | | | |
| | Pr(L) | p | Pr(M-L) | p | Pr(M-H) | p | Pr(H) | p |
| Age | 0.10 [−0.00, 0.20] | .06 | −0.03 [−0.15, 0.09] | .62 | −0.06 [−0.18, 0.05] | .28 | −0.00 [−0.06. 0.06] | .91 |
| Gender: Men | 0.11 [−0.22, 0.45] | .51 | −0.06 [−0.49, 0.36] | .78 | −0.12 [−0.50, 0.26] | .54 | 0.07 [0.01, 0.12] | .01 |
| University or higher | 0.14 [−0.07, 0.35] | .19 | 0.08 [−0.15, 0.31] | .47 | −0.12 [−0.32, 0.08] | .22 | −0.10 [−0.19, −0.02] | .02 |
| Living alone | −0.11 [−0.33, 0.11] | .32 | 0.06 [−0.17, 0.29] | .63 | −0.04 [−0.24, 0.16] | .69 | 0.10 [0.04, 0.15] | .002 |
| Comorbidities (count) | −0.03 [−0.06, 0.00] | .09 | 0.01 [−0.03, 0.04] | .68 | 0.03 [0.00, 0.06] | .03 | −0.01 [−0.03, 0.01] | .22 |
| Self-mastery | 0.35 [0.05, 0.64] | .02 | 0.05 [−0.28, 0.37] | .79 | −0.09 [−0.38, 0.21] | .57 | −0.31 [−0.56, −0.06] | .02 |
| HIV stigma | 0.03 [−0.02, 0.08] | .25 | −0.01 [−0.06, 0.05] | .83 | 0.02 [−0.03, 0.07] | .52 | −0.04 [−0.07, −0.01] | .02 |
| Social support | 0.03 [−0.02, 0.07] | .24 | −0.02 [−0.06, 0.03] | .52 | −0.01 [−0.05, 0.03] | .77 | −0.01 [−0.02, 0.01] | .49 |
| | **Uncertainty** | | | | | | | |
| Contextual factors | Pr(L) | p | Pr(M-L) | p | Pr(M-H) | p | Pr(H) | p |
| Age | 0.03 [−0.04, 0.10] | .38 | −0.10 [−0.20, 0.00] | .05 | 0.07 [−0.01, 0.15] | .07 | −0.00 [−0.00, 0.00] | .44 |
| Gender: Men | 0.13 [−0.06, 0.33] | .18 | −0.02 [−0.36, 0.33] | .92 | −0.06 [−0.36, 0.23] | .68 | −0.05 [−0.18, 0.07] | .39 |
| University or higher | 0.08 [−0.08, 0.24] | .32 | 0.04 [−0.18, 0.25] | .74 | −0.02 [−0.19, 0.15] | .81 | −0.10 [−0.15, −0.05] | <.001 |
| Living alone | 0.07 [−0.09, 0.22] | .40 | 0.20 [0.00, 0.39] | .04 | −0.14 [−0.27, −0.00] | .05 | −0.13 [−0.13, −0.12] | <.001 |
| Comorbidities (count) | −0.03 [−0.05, −0.01] | .003 | 0.04 [0.01, 0.06] | .01 | −0.00 [−0.02, 0.02] | .69 | −0.00 [−0.00, 0.00] | .75 |
| Self-mastery | 0.07 [−0.16, 0.30] | .57 | 0.33 [0.04, 0.63] | .03 | −0.40 [−0.62, −0.17] | <.001 | −0.00 [−0.01, 0.01] | .57 |
| HIV stigma | −0.06 [−0.09, −0.03] | <.001 | 0.03 [−0.02, 0.07] | .26 | 0.03 [−0.00, 0.06] | .08 | 0.00 [−0.01, 0.01] | .57 |
| Social support | 0.04 [0.01, 0.06] | .01 | −0.02 [−0.06, 0.02] | .32 | −0.02 [−0.05, 0.02] | .36 | −0.00 [−0.00, 0.00] | .58 |
| | **Challenges to social inclusion** | | | | | | | |
| Contextual factors | Pr(L) | p | Pr(M-L) | p | Pr(M-H) | p | Pr(H) | p |
| Age | 0.02 [−0.07, 0.10] | .39 | 0.05 [−0.02, 0.13] | .17 | −0.05 [−0.18, 0.08] | .48 | −0.02 [−0.12, 0.07] | .63 |
| Gender: Men | −0.34 [−0.76, 0.08] | .11 | 0.18 [0.10, 0.27] | <.001 | 0.22 [−0.21, 0.65] | .32 | −0.06 [−0.31, 0.18] | .62 |
| University or higher | 0.03 [−0.16, 0.22] | .73 | 0.00 [−0.17, 0.18] | .98 | −0.07 [−0.30, 0.17] | .57 | 0.03 [−0.11, 0.17] | .66 |
| Living alone | 0.12 [−0.06, 0.29] | .20 | −0.10 [−0.28, 0.09] | .30 | −0.05 [−0.28, 0.18] | .67 | 0.03 [−0.11, 0.18] | .67 |
| Comorbidities (count) | −0.02 [−0.04, 0.01] | .29 | −0.03 [−0.06, −0.00] | .05 | 0.03 [−0.00, 0.06] | .09 | 0.02 [−0.00, 0.03] | .07 |
| Self-mastery | −0.02 [−0.31, 0.27] | .91 | 0.19 [−0.07, 0.45] | .15 | −0.16 [−0.47, 0.16] | .33 | −0.01 [−0.19, 0.17] | .87 |
| HIV stigma | −0.05 [−0.09, −0.01] | .02 | −0.00 [−0.04, 0.04] | .92 | 0.01 [−0.04, 0.06] | .58 | 0.04 [0.01, 0.07] | .01 |
| Social support | 0.05 [0.01, 0.08] | .01 | −0.00 [−0.04, 0.03] | .90 | −0.03 [−0.07, 0.02] | .24 | −0.02 [−0.04, 0.00] | .11 |

Marginal mean probabilities were calculated from the multinomial logistic regression models. Pr() = change in the probability of being allocated to a certain trajectory group. The 95% confidence intervals were presented in square brackets. To facilitate interpretation, we reported the results for contextual factors, including age, self-mastery, HIV stigma scores, and social support, scaled by a factor of 10 to interpret the change in predicted probabilities in terms of a decade increase or a 10-point increase in these factors.

## Discussion

We characterized disability trajectories across six different dimensions and investigated the influence of contextual factors among adults aging with HIV over an eight-month period. We identified three distinct disability trajectories (low, medium, high) for physical symptoms, mental-emotional symptoms, and difficulties with day-to-day activities, and four trajectories (low, medium-low, medium-high, high) for cognitive symptoms, uncertainty, and challenges to social inclusion. Modifiable contextual factors such as self-mastery and social support were associated with lower disability trajectories, whereas comorbidities and HIV stigma were associated with more severe disability trajectories over time. The present study underscores the complex nature of disability and its diverse influencing factors among adults aging with HIV.

### Experiences of disability in the context of aging with HIV

The mean participant age of 50.6 years is similar to the mean age reported in a population-based study of people living with HIV in Ontario, Canada (51.3±12.3 years) [42]. Additionally, participants had lived with HIV for an average of 17.5±9.9 years, which reflects prolonged exposure to the health-related challenges associated with aging with HIV. A growing body of evidence has documented the intertwined relationship between HIV and disability [43–46]. For example, Brown and colleagues reported that the prevalence of moderate disability, as measured by the generic World Health Organization Disability Assessment Schedule 2.0 (WHODAS), was over 70% among adults living with HIV who access routine outpatient HIV care in London, United Kingdom (UK) [47]. More broadly, a systematic review of 61 studies demonstrated that the experience of disability, defined as any functional impairments, activity limitations, or participation restrictions, was prevalent among individuals living with HIV in sub-Saharan Africa [45]. Our study contributes to the literature by adopting the SF-HDQ, an established patient-reported outcome measure, to assess the multidimensional nature of disability in the context of living with HIV over time. Our findings reveal that, even among a group of community-dwelling adults aging with HIV who are medically stable and able to conduct exercise, some individuals may still experience higher levels of disability than others over the course of living with HIV. In line with notions from prior research, the distinct disability dimension trajectories identified in this study highlight the necessity of adopting a tailored, individualized approach rather than viewing people aging with HIV as a single homogenous group [15,19,22,48]. In addition, analogous to the concept of triage in medical practice, recognizing these diverse experiences can help identify individuals with greater needs and guide healthcare professionals in refining treatments and optimizing the provision of person-centred services for people aging with HIV.

### Stability and variability of disability trajectories

Overall, the mean trajectories of disability were observed to remain stable over the eight-month timeframe. However, it is important to underline the fluctuations and variations in disability trajectories across the individual participants assigned to each trajectory group. For instance, upon closer examination of the physical symptoms dimension, it became evident that there were notable within-person fluctuations in the severity of disability for each study participant (Fig 1). Moreover, even within the same trajectory group, we noted between-person differences in the experience of disability over time. While we employed domain-specific scores to measure disability, our findings are consistent with Solomon et al.'s work, where they also discovered similar fluctuations in the levels of disability among individuals aging with HIV using item-specific questions and a qualitative longitudinal study design [48]. To help contextualize the findings, we presented the "good day/bad day" item on the SF-HDQ and found that more than a third of participants reported a mixture of good days and bad days during the timeline, suggesting that adults aging with HIV can experience fluctuations in their health coupled with varying levels of disability over time. As outlined in the *Episodic Disability Framework*, these observed patterns exemplify the characteristics of episodic disability, where individuals may experience fluctuating periods of illness and wellness, a phenomenon documented not only in HIV, but also other chronic health conditions such as Long COVID, arthritis, and relapsing-remitting multiple sclerosis [49–51]. Altogether, this study portrays the episodic nature and complexity of disability among individuals aging with HIV, stressing the importance of continuous monitoring and developing interventions that address the specific health challenges encountered by this population.

### Clinical distinction of disability trajectories

Results from this study identified three to four distinct trajectories in each disability dimension. These trajectories, with SF-HDQ scores ranging from 3 to 55 on the 100-point scale, were considered statistically distinct using a data-driven approach. From a clinical perspective however, the decision regarding whether three or four trajectories should be adopted requires further exploration. This is particularly relevant because clinically meaningful differences in the SF-HDQ

scores between groups are not yet established for each of the six disability dimensions. Considering the precision and interpretability of the SF-HDQ instrument [27], it is plausible that the actual number of clinically distinct trajectory groups may be fewer than identified in this analysis. Further verification is thus needed to determine the extent to which the observed differences are clinically important, beyond associated measurement error, and have practical significance. Collectively, the evolving interpretability of the SF-HDQ should be taken into consideration by researchers and clinicians when evaluating the results and developing subsequent intervention strategies for individuals aging with HIV.

### Influence of contextual factors on disability trajectories

Investigators have identified risk factors for disability among people living with HIV. For example, length of time since HIV diagnosis, socioeconomics status, and use of rehabilitation services were cross-sectionally associated with disability, as measured by both WHODAS and the HDQ, among people living with HIV in the UK [47]. Our study adds to the extant literature by revealing how intrinsic and extrinsic contextual factors influence disability trajectories. Our findings highlight that beyond individual characteristics such as age, gender, or education, having a greater number of comorbidities and higher levels of stigma at baseline were associated with more severe disability trajectories for the physical, cognitive, uncertainty, difficulty with day-to-day activities, and challenges to social inclusion disability dimensions. On the contrary, greater baseline self-mastery and perceived social support were associated with lower trajectories (indicating better health) in these disability dimensions over time.

The observed deleterious impact of comorbidities and stigma, along with the protective role of self-mastery and social support, aligns with existing research [52–56] and helps to identify people living with HIV who may be more susceptible to greater disability, providing insights for shaping future care and services. Specifically, our findings emphasize the importance of adopting an integrated approach to support people aging with HIV, with the focus extending beyond standard HIV treatment to address other comorbidities such as mental health conditions and musculoskeletal disorders. In the meantime, opportunities may also be provided to enhance mastery, coping skills, and sense of control among people managing chronic health conditions such as HIV [54,57,58]. Moreover, our study highlights the critical role of building social support systems in the community, while emphasizing the urgent need for broader structural changes to combat HIV-related stigma and discrimination [56,59,60]. By drawing attention across various entities, from clinical care to public health, our study highlights the collective efforts needed to support the health and well-being of people living with HIV.

### Strengths and limitations

Strengths of our approach included our use of the *Episodic Disability Framework*, which offers a robust conceptual foundation for capturing the multidimensional and episodic nature of disability and contextual factors in the context of HIV [13]. This study is among the first known to employ a validated HIV-specific questionnaire that uses a Rasch-based interval scale to assess disability longitudinally over time [24]. However, several limitations exist. First, the sample size is smaller, and the follow-up period is shorter compared with previous studies that used public survey data to examine disability trajectories among general older populations (e.g., studies with hundreds or thousands of respondents) [61–63]. The numbers and patterns of disability trajectories might have changed with a larger cohort and longer follow-up duration (e.g., years). Second, participants were community dwelling adults recruited from HIV community-based organizations in Toronto, who were predominantly men, White, and medically stable. Given the characteristics of the participants in this study, findings may not be transferable to individuals of different genders, racial/ethnic backgrounds, and those living in rural areas or with diverse sociodemographic characteristics. Third, the current study assessed the role of contextual factors at baseline, whereas these factors may also change over time and influence disability trajectories. Furthermore, although our study serves as the first step in adopting a multidimensional approach to examine disability trajectories across various life domains, the intersectionality of these dimensions exceeds our current scope [64]. An important

avenue for future research is to leverage additional analytical procedures, such as group-based multi-trajectory modeling, to characterize the correlation or latent clusters of individuals exhibiting similar trajectories across all six disability dimensions. In conjunction with these techniques, the application of other numeric metrics such as intra-individual mean (iMean), intra-individual standard deviation (iSD), and entropy may provide further insights into the dynamics and episodic nature of disability over time [65].

In conclusion, our findings suggest that experiences of disability among adults aging with HIV included three or four distinct trajectories over an eight-month monitoring phase. The heterogeneity in disability trajectories underscores the need for personalized care. Targeting modifiable factors like stigma (e.g., through anti-stigma campaigns) and bolstering self-mastery and social support represent promising avenues for interventions designed to mitigate disability among adults aging with HIV.

## Supporting information

**S1 Fig. Overview of study design and measurement timepoints during the baseline monitoring phase of the community-based exercise (CBE) intervention study.**
(PDF)

**S2 Fig. The interconnections between trajectories across the six disability dimensions.**
(PDF)

**S1 Table. Characteristics of participants who initiated and remained at the end of baseline monitoring phase (n = 83).**
(PDF)

**S2 Table. Trajectory coefficients for physical, mental-emotional symptoms, and day-to-day activity difficulties.**
(PDF)

**S3 Table. Trajectory coefficients for cognitive symptoms, uncertainty, and challenges to social inclusion.**
(PDF)

**S4 Table. Average posterior probability of group assignment.**
(PDF)

## Author contributions

**Conceptualization:** Tai-Te Su, Ahmed M. Bayoumi, Soo Chan Carusone, Ada Tang, Patricia Solomon, Aileen M. Davis, Kelly K. O'Brien.

**Data curation:** Ahmed M. Bayoumi, Soo Chan Carusone, Ada Tang, Patricia Solomon, Aileen M. Davis, Kelly K. O'Brien.

**Formal analysis:** Tai-Te Su.

**Funding acquisition:** Ahmed M. Bayoumi, Soo Chan Carusone, Ada Tang, Patricia Solomon, Aileen M. Davis, Kelly K. O'Brien.

**Investigation:** Tai-Te Su, Ahmed M. Bayoumi, Lisa Avery, Soo Chan Carusone, Ada Tang, Patricia Solomon, Aileen M. Davis, Kelly K. O'Brien.

**Methodology:** Tai-Te Su, Ahmed M. Bayoumi, Lisa Avery, Soo Chan Carusone, Ada Tang, Patricia Solomon, Aileen M. Davis, Kelly K. O'Brien.

**Project administration:** Kelly K. O'Brien.

**Resources:** Ahmed M. Bayoumi, Lisa Avery, Soo Chan Carusone, Ada Tang, Patricia Solomon, Aileen M. Davis, Kelly K. O'Brien.

**Software:** Kelly K. O'Brien.

**Supervision:** Kelly K. O'Brien.

**Visualization:** Tai-Te Su.

**Writing – original draft:** Tai-Te Su.

**Writing – review & editing:** Tai-Te Su, Ahmed M. Bayoumi, Lisa Avery, Soo Chan Carusone, Ada Tang, Patricia Solomon, Aileen M. Davis, Kelly K. O'Brien.

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
