## [Decision Letter · Decision Letter 0]

4 Mar 2025

Dear Dr. O'Brien,

Thank you for submitting your manuscript to PLOS ONE. After careful consideration, we feel that it has merit but does not fully meet PLOS ONE’s publication criteria as it currently stands. Therefore, we invite you to submit a revised version of the manuscript that addresses the points raised during the review process.

We look forward to receiving your revised manuscript.

Kind regards,

Xiangwei Li

Academic Editor

PLOS ONE

2. We note that your Data Availability Statement is currently as follows: [All relevant data are within the paper and its Supporting information files.]

Reviewer's Responses to Questions

**Comments to the Author**

1. Is the manuscript technically sound, and do the data support the conclusions?

Reviewer #1: Yes

Reviewer #2: Yes

2. Has the statistical analysis been performed appropriately and rigorously?

Reviewer #1: Yes

Reviewer #2: Yes

3. Have the authors made all data underlying the findings in their manuscript fully available?

Reviewer #1: Yes

Reviewer #2: No

4. Is the manuscript presented in an intelligible fashion and written in standard English?

Reviewer #1: Yes

Reviewer #2: Yes

Reviewer #1: Thank you to the authors for this considered and comprehensive manuscript that will add value to the field of HIV disability and rehabilitation. My feedback is provided below:

1) Line 91 data point 0.32, this may not be fully intuitive to the reader therefor may require either some context or explanation of what 0.32 additional functional limitation means.

2) Line 97 single aspect of disability may not be understood by some readers, therefore please consider expanding eg: only mental and emotional impairments.

3) Lines 142-143, the single summary score may not be known as an original feature of the full HIV Disability Questionnaire (HDQ), therefore it is unclear why this information is included in the description of the SF-HDQ. Please can you explain what the relevance of not having tyhe summary score is, to rationalise including this sentence.

4) Figure 1: There is not consistency in group presentation eg: [A] presents in order medium, low, high, but [C] presents low, medium, high. Can all figures in figure 1 be presented in the same order from low to high as described in lines 256-259?

5) Are high and high-declining trajectories (line 258) the same or different, as high implies a stable trajectory compared to high-declining.

6) I am unclear from the trajectories describe over 8 months, if this dopes or does not include the episodic or fluctuating nature of disability over this time frame. In the results (lines 267-287), can people transition between trajectories?

7) For participants who reported good or bad days, is the data presented for bad days at any time point? (line 286) and does this reflect an episodic change during a trajectory?

8) I am not clear on the significance or value of 3 versus 4 trajectories between differing dimensions (lines 340-344). Can this be discussed eg: in discussion paragraph lines 386-395?

9) Lines 362-364, could the concept of "triaging" trajectories assist in describing the provision of person-centred care.

10) Line 353 there is the opportunity to include additional disability prevalence data eg: https://journals.plos.org/plosone/article?id=10.1371/journal.pone.0267271

11) The discussion of associations between extrinsic and intrinsic conceptual factors and disability severity across dimensions would benefit from including data form existing literature eg: "Participants who were diagnosed with HIV late, were economically inactive, received benefits, and received rehabilitation in past 12-months were associated with statistically significant increased odds for severe disability" https://journals.plos.org/plosone/article?id=10.1371/journal.pone.0267271

12) Grammar suggestion (line 446) please consider adjusting to "interventions, services and policies aimed at preventing or mitigating disability to improve health health outcomes among adults aging with HIV"

Reviewer #2: In this manuscript, the authors address a crucial gap in the literature looking at disability trajectories among older adults living with HIV, using a multidimensional definition for disability. Overall the manuscript is well written and technologically sound. The authors incorporate several important contextual factors (both modifiable and non-modifiable) to help contextualize differences in disability trajectories overtime. The findings are limited by a relatively short follow up period (8 months) which may explain why most of the "trajectories" are stable over time.

A few considerations are outlined below to help improve the clarity of the work:

1. Ensure there is consistency in how things are labeled in Table 1 (i.e. gender is follow by (n) but race/ethnicity is not) and organized in Figure 1 (i.e. use the same order for presenting trajectory groups across domains such as Low, followed by Medium, followed by High).

2. In the results section where authors describe % of participants in various disability groups (starting on line 267) it is a bit confusing when certain groups are being referred to as "consistently low level of disability" (line 268) vs others as "low disability trajectory" (line 275) when both trajectories look very similar on in Figure 1. It makes it sound like there is a difference between the two (the former staying the same over time while the latter declining). Perhaps using the same language consistently would help avoid this confusion, i.e. calling both trajectories.

3. Was there a correlation between disability trajectory group assignment across domains of disability? For example, are those assigned to the high physical symptoms trajectory group more likely to also be in the high challenges to social inclusion trajectory?

4. How was good vs. bad day assignment taken into account in the analysis?

what does this mean?

---

## [Author Response · Author response to Decision Letter 1]

31 Mar 2025

Review Comments to the Author:

Reviewer #1

Thank you to the authors for this considered and comprehensive manuscript that will add value to the field of HIV disability and rehabilitation. My feedback is provided below:

1) Line 91 data point 0.32, this may not be fully intuitive to the reader therefor may require either some context or explanation of what 0.32 additional functional limitation means.

RESPONSE-1: Thank you for providing these positive comments and feedback. We revised our reporting of the literature to improve clarity. The sentence in the Introduction now reads: “For instance, Crystal and Sambamoorthi identified an increase in the number of basic daily activities that individuals living with HIV had difficulty performing independently (0.32 additional tasks per month) over a two-year follow-up period [15].” (p.5, lines 89-91)

2) Line 97 single aspect of disability may not be understood by some readers, therefore please consider expanding eg: only mental and emotional impairments.

RESPONSE-2: We have re-written the sentence as follows “While this evidence provides critical insights, each of these studies focused on a single dimension of disability, such as difficulties with day-to-day activities or mental or emotional symptoms, as outlined in the Episodic Disability Framework.” (p.6, lines 95-98)

3) Lines 142-143, the single summary score may not be known as an original feature of the full HIV Disability Questionnaire (HDQ), therefore it is unclear why this information is included in the description of the SF-HDQ. Please can you explain what the relevance of not having the summary score is, to rationalise including this sentence.

RESPONSE-3: We added a rationale for why a single summary score is not available in the short-from HIV Disability Questionnaire (SF-HDQ). We revised the manuscript to include the following sentence: “A single summary score for disability is not provided by the SF-HDQ due to the unidimensionality assumption of the Rasch model and the distinct nature of each disability dimension.” (p8, lines 142-144)

4) Figure 1: There is not consistency in group presentation eg: [A] presents in order medium, low, high, but [C] presents low, medium, high. Can all figures in figure 1 be presented in the same order from low to high as described in lines 256-259?

RESPONSE-4: We agree that it is important to remain consistent in reporting trajectories across disability dimensions. We revised Figure 1 so that the trajectory groups across all six disability dimensions follow the suggested order (from low to high).

5) Are high and high-declining trajectories (line 258) the same or different, as high implies a stable trajectory compared to high-declining.

RESPONSE-5: Thank you for raising this crucial point and the opportunity to clarify. The term “high-declining trajectory” was originally used for the cognitive symptom dimension considering the statistically significant intercept (b = 74.21, SE = 5.00, p < .001) and the negative linear slope (b = -0.30, SE = 0.10, p < .01). Additionally, the quadratic slope was positive and statistically significant (b = 0.00, SE = 0.00, p < .05; values were shown as 0 due to rounding), suggesting a minor upward curvature. Upon careful consideration of the reviewer’s question, we renamed the “high-declining” trajectory to “high” to maintain consistency in trajectory naming and minimize potential confusion for readers. We deem that this change is justified given that the overall level of disability remains high over time and aligns with our other high-intercept trajectories.

6) I am unclear from the trajectories described over 8 months, if this does or does not include the episodic or fluctuating nature of disability over this time frame. In the results (lines 267-287), can people transition between trajectories?

RESPONSE-6: Thank you for this thoughtful comment. In our analysis, we identified distinct disability trajectories across six dimensions among adults aging with HIV over an 8-month period. While our findings reflect group-level differences (e.g., low vs. medium vs. high), we also observed considerable variability among individuals within the same trajectory group (as illustrated in Figure 1). This variability supports the notion that disability is episodic and fluctuating in nature, as described in the Episodic Disability Framework. We discussed these in detail in the Discussion section of the revised manuscript (p.24, lines 372-393).

Regarding trajectory group membership, we included the following sentence to clarify this point: “For each disability dimension, participants were assigned to one specific trajectory group based on the highest posterior probability and thus their group membership remained unchanged over the 8-month timeframe.” (p.15, lines 268-270)

7) For participants who reported good or bad days, is the data presented for bad days at any time point? (line 286) and does this reflect an episodic change during a trajectory?

RESPONSE-7: Thank you, we revised the sentence to clarify the reporting of good and bad days as follows: “Throughout the 8-month monitoring phase, 60.2% of participants consistently reported having ‘good days’ when they completed the SF-HDQ, while 1.9% consistently reported ‘bad days’ across all bimonthly assessments. Approximately 38% reported experiencing both good and bad days at different time points during this period.” Additionally, we expanded our discussion on the interpretations of self-reported experiences of good and bad days in relation to disability trajectories in the revised manuscript (p.24, lines 382-386)

8) I am not clear on the significance or value of 3 versus 4 trajectories between differing dimensions (lines 340-344). Can this be discussed eg: in discussion paragraph lines 386-395?

RESPONSE-8: In this study the optimal number of trajectories in each disability dimension was determined based on the parsimony principal and a combination of model fit statistics. While the identified trajectories were considered statistically distinct, the extent to which these differences are meaningful in clinical practice requires further investigation. To incorporate the reviewer’s comment, we added the following sentences to the “Clinical distinction of disability trajectories” section in the revised manuscript: “From a clinical perspective however, the decision regarding whether three or four trajectories should be adopted requires further exploration. This is particularly relevant because clinically meaningful differences in the SF-HDQ scores between groups are not yet established for each of the six disability dimensions.” (p.25, lines 397-400)

9) Lines 362-364, could the concept of "triaging" trajectories assist in describing the provision of person-centred care.

RESPONSE-9: We agree and incorporated the concept of triaging in the revised Discussion section: “In addition, analogous to the concept of triage in medical practice, recognizing these diverse experiences can help identify individuals with greater needs and guide healthcare professionals in refining treatments and optimizing the provision of person-centred services for people aging with HIV.” (p.24, lines 368-371)

10) Line 353 there is the opportunity to include additional disability prevalence data eg: https://journals.plos.org/plosone/article?id=10.1371/journal.pone.0267271

RESPONSE-10: As suggested we included: “For example, Brown and colleagues reported that the prevalence of moderate disability, as measured by the generic World Health Organization Disability Assessment Schedule 2.0 (WHODAS), was over 70% among adults living with HIV who access routine outpatient HIV care in London, United Kingdom [46].” (p.23, lines 354-357)

11) The discussion of associations between extrinsic and intrinsic conceptual factors and disability severity across dimensions would benefit from including data form existing literature eg: "Participants who were diagnosed with HIV late, were economically inactive, received benefits, and received rehabilitation in past 12-months were associated with statistically significant increased odds for severe disability" https://journals.plos.org/plosone/article?id=10.1371/journal.pone.0267271

RESPONSE-11: We added the following sentence to the Discussion section: “Investigators have identified risk factors for disability among people living with HIV. For example, length of time since HIV diagnosis, socioeconomics status, and use of rehabilitation services were cross-sectionally associated with disability, as measured by both WHODAS and the HDQ, among people living with HIV in the UK [46].” (p.25, lines 409-412)

12) Grammar suggestion (line 446) please consider adjusting to "interventions, services and policies aimed at preventing or mitigating disability to improve health outcomes among adults aging with HIV"

RESPONSE-12: We revised the sentence as suggested: “A combination of modifiable contextual factors such as comorbidities, self-mastery, stigma, and social support may influence disability trajectories over time and should be incorporated into interventions, services, and policies aimed at preventing or mitigating disability to improve health outcomes among adults aging with HIV.” (p.28, lines 458-462)

Reviewer #2

In this manuscript, the authors address a crucial gap in the literature looking at disability trajectories among older adults living with HIV, using a multidimensional definition for disability. Overall the manuscript is well written and technologically sound. The authors incorporate several important contextual factors (both modifiable and non-modifiable) to help contextualize differences in disability trajectories overtime. The findings are limited by a relatively short follow up period (8 months) which may explain why most of the "trajectories" are stable over time.

A few considerations are outlined below to help improve the clarity of the work:

1) Ensure there is consistency in how things are labeled in Table 1 (i.e. gender is follow by (n) but race/ethnicity is not) and organized in Figure 1 (i.e. use the same order for presenting trajectory groups across domains such as Low, followed by Medium, followed by High).

RESPONSE-1: Thank you for this observation and thoughtful feedback. We carefully reviewed and made the corrections in Table 1. Additionally, in response to both reviewers’ suggestions, we revised Figure 1. to ensure that trajectory groups across all disability dimensions follow a consistent order and colour scheme.

2) In the results section where authors describe % of participants in various disability groups (starting on line 267) it is a bit confusing when certain groups are being referred to as "consistently low level of disability" (line 268) vs others as "low disability trajectory" (line 275) when both trajectories look very similar on in Figure 1. It makes it sound like there is a difference between the two (the former staying the same over time while the latter declining). Perhaps using the same language consistently would help avoid this confusion, i.e. calling both trajectories.

RESPONSE-2: We agree that the previous wording may have caused confusion. We revised the sentences to use consistent terminology when describing trajectories across all six disability dimensions. The revisions are marked in red on p.15 between lines 268-277.

3) Was there a correlation between disability trajectory group assignment across domains of disability? For example, are those assigned to the high physical symptoms trajectory group more likely to also be in the high challenges to social inclusion trajectory?

RESPONSE-3: Thank you for this insightful question. In the present study we examined disability trajectories in each of the six dimensions separately using group-based trajectory modeling. As a result, assessing correlations between trajectory group assignments across dimensions was beyond the scope of our analysis. However, in response to the reviewer’s question, we generated a new supplemental file with a Sankey diagram (S1 Fig) to visually illustrate the interconnections between trajectories identified in each disability dimension. Additionally, we highlighted in the Strengths and Limitations section that future research may consider employing advanced techniques, such as group-based multi-trajectory modeling (an extension of the method used in our study), to formally investigate these relationships.

4) How was good vs. bad day assignment taken into account in the analysis?

RESPONSE-4: In this study we followed established recommendations and estimated the disability trajectories as a function of time only to avoid potential contamination of group classification. However, we presented findings on good days and bad days to provide additional context for interpreting the results. We clarified this in the Discussion section as follows: “To help contextualize the findings, we presented the “good day/ bad day” item on the SF-HDQ and found that more than a third of participants reported a mixture of good days and bad days during the timeline, suggesting that adults aging with HIV can experience fluctuations in their health coupled with varying levels of disability over time.” (p.24, line 382-386)

---

## [Decision Letter · Decision Letter 1]

30 Oct 2025

Dear Dr. O'Brien,

Thank you for submitting your manuscript to PLOS ONE. After careful consideration, we feel that it has merit but does not fully meet PLOS ONE’s publication criteria as it currently stands. Therefore, we invite you to submit a revised version of the manuscript that addresses the points raised during the review process.

We look forward to receiving your revised manuscript.

Kind regards,

Stanley Chinedu Eneh

Academic Editor

PLOS ONE

**Journal Requirements:**

Reviewers' comments:

Reviewer's Responses to Questions

**Comments to the Author**

Reviewer #1: All comments have been addressed

Reviewer #3: All comments have been addressed

2. Is the manuscript technically sound, and do the data support the conclusions?

Reviewer #1: Yes

Reviewer #3: Yes

3. Has the statistical analysis been performed appropriately and rigorously?

Reviewer #1: Yes

Reviewer #3: Yes

4. Have the authors made all data underlying the findings in their manuscript fully available?

Reviewer #1: Yes

Reviewer #3: Yes

5. Is the manuscript presented in an intelligible fashion and written in standard English?

Reviewer #1: Yes

Reviewer #3: Yes

**Reviewer #1: ** All comments via peer review have been fully considered and addressed by the study authors. The paper is suitable for publication and will add value to the literature base in the field of HIV, disability and rehabilitation.

**Reviewer #3: ** This is a strong and good manuscript that effectively communicates the purpose, methods, key findings, and implications of the research. It is well-suited for a scientific audience and clearly demonstrates the value of the study. The methodology is sophisticated and appropriate for the research questions.

By incorporating more descriptive language for the trajectories and using stronger, more causal language to describe the associations, the authors can elevate an already excellent abstract to be even more compelling and informative.

**Do you want your identity to be public for this peer review?** For information about this choice, including consent withdrawal, please see our Privacy Policy

Reviewer #1: No

Reviewer #3: No

---

## [Author Response · Author response to Decision Letter 2]

11 Nov 2025

Review Comments to the Author:

Editor

1) I have evaluated the original and revised manuscript(s) and can confirm that your methodology is sound. However, I notice that the current manuscript is closely linked to O'Brien et al. (2016) https://doi.org/10.1136/bmjopen-2016-013618. To further strengthen your methods section, I recommend including a flow chart that illustrates the data source, showing how data were collected, the five time points, and the measures used. This will make the methodology clearer and more accessible to readers.

RESPONSE-1: Thank you for the positive feedback and suggestion. We developed a flow chart illustrating the study design, data sources, five measurement timepoints, and corresponding measures from the community-based exercise (CBE) study. We added this figure to the revised manuscript as Supplementary Figure S1 (Page 7, lines 134-135).

Reviewer #1

1) All comments via peer review have been fully considered and addressed by the study authors. The paper is suitable for publication and will add value to the literature base in the field of HIV, disability and rehabilitation.

RESPONSE-1: We thank the reviewer for providing these positive comments. We appreciate your valuable suggestions and recognition of our investigation.

Reviewer #3

This is a strong and good manuscript that effectively communicates the purpose, methods, key findings, and implications of the research. It is well-suited for a scientific audience and clearly demonstrates the value of the study. The methodology is sophisticated and appropriate for the research questions. By incorporating more descriptive language for the trajectories and using stronger, more causal language to describe the associations, the authors can elevate an already excellent abstract to be even more compelling and informative.

1) The abstract states that trajectories were found but doesn't describe their nature. For example, what does "high-declining" mean in the social inclusion dimension? Did it improve or worsen? A brief descriptor for the key trajectories (e.g., "a 'high-stable' trajectory," "a 'low-improving' trajectory") would add significant meaning.

RESPONSE-1: Thank you for the thoughtful observation and opportunity to clarify. The identified disability trajectories primarily differed in their baseline levels (intercepts) and remained relatively stable over time. To improve clarity and ensure consistent terminology across dimensions, we revised the label “high-declining” to “high” in the cognitive symptom dimension during the previous round of review (R1). This revision is reflected in the current manuscript. We discussed the implications of these findings in the Discussion section (Page 24, lines 379-400).

2) The results for the multinomial regression are vague. Stating that factors "were associated with" trajectories is weak. A more impactful statement would specify the direction and, if space allows, the strongest predictors (e.g., "Higher stigma was the strongest predictor of belonging to the 'high-severity' physical disability trajectory").

RESPONSE-2: Thank you for the insightful comment. In our analyses, we used information on contextual factors measured at baseline. Thus, while associations can be identified, we were unable to strictly determine causal direction, which we acknowledged as a limitation (Page 27, lines 454-456). Because each contextual factor was measured on different scales (e.g., continuous for age and categorical for gender), direct comparison of effect magnitudes is challenging. Hence, we focused on describing the direction and statistical significance of each association when interpreting the findings.

3) The sample is overwhelmingly male (89%). While this may reflect the epidemiology of the aging HIV population in the study context, it is a limitation that should be briefly acknowledged, as the findings may not be generalizable to women aging with HIV.

RESPONSE-3: Following the suggestion, we revised the Limitations section to acknowledge the limited gender representation in our sample. The revised sentence now reads: “Given the characteristics of the participants in this study, findings may not be transferable to individuals of different genders, racial/ethnic backgrounds, and those living in rural areas or with diverse sociodemographic characteristics.” (Page 27, lines 451-454)

4) The mean age of 50.6 is relatively young for "aging" in many gerontological studies. A brief justification for this age cutoff (e.g., the premature aging observed in HIV) would be helpful for a broader audience.

RESPONSE-4: Thank you for this thoughtful comment. We did not apply an age cutoff to define aging and instead conceptualized it as a process that unfolds over time while living with a complex chronic condition such as HIV. In our sample, participants had lived with HIV for an average of 17.5±9.9 years, reflecting prolonged exposure to the health, functional, and social challenges associated with aging with HIV over the lifespan. The mean age of 50.6 years in our study is similar to the mean age reported in a population-based study of people living with HIV in Ontario, Canada (mean = 51.3 years; Coelho et al., 2025). In response to your suggestion, we included the corresponding sentences in the revised manuscript (Page 23, lines 356-359).

5) The number of participants (n=108) is modest for a longitudinal study with 5 time points and a complex statistical model (LCGA). While not necessarily a flaw, it's a contextual factor that a critical reader would note. The authors are likely confident in their model fit, but it's an inherent limitation of the study design.

RESPONSE-5: We clarified sample size as a limitation in the revised manuscript as follows: “First, the sample size is smaller, and the follow-up period is shorter compared with previous studies that used public survey data to examine disability trajectories among general older populations (e.g., studies with hundreds or thousands of respondents) [61–63].” (Page 27, lines 445-448)

6) It is unclear if the contextual factors were measured only at baseline (which is implied) or also longitudinally. Using only baseline measures is common, but it limits the ability to model how changes in social support or stigma, for example, might influence disability trajectories.

RESPONSE-6: We followed the suggestion and clarified the timing of the measurement of contextual factors in the Methods section: “After participants were assigned to their respective trajectory group in each of the six disability dimensions, we conducted multinomial logistic regression models to assess the influence of intrinsic contextual factors (age, gender, education, living status, number of comorbidities, mastery) and extrinsic contextual factors (stigma, social support) measured at baseline on subsequent disability trajectories.” (Page 11, lines 212-216)

7) Suggested Strengthened Conclusion: "The heterogeneity in disability trajectories underscores the need for personalized care. Targeting modifiable factors like stigma (e.g., through anti-stigma campaigns) and bolstering self-mastery and social support represent promising avenues for interventions designed to mitigate disability among adults aging with HIV."

RESPONSE-7: Thank you. We incorporated the suggested strengthened conclusion in the Conclusion section of the main manuscript (Page 28, lines 466-470).

---

## [Editor Report · Decision Letter 2]

19 Nov 2025

Trajectories of disability and influence of contextual factors among adults aging with HIV: insights from a community-based longitudinal study in Toronto, Canada

PONE-D-24-33533R2

Dear Dr. O'Brien,

We’re pleased to inform you that your manuscript has been judged scientifically suitable for publication and will be formally accepted for publication once it meets all outstanding technical requirements.

Kind regards,

Stanley Chinedu Eneh

Academic Editor

PLOS ONE
---

## [Editor Report · Acceptance letter]

PONE-D-24-33533R2

PLOS ONE

Dear Dr. O'Brien,

I'm pleased to inform you that your manuscript has been deemed suitable for publication in PLOS ONE. Congratulations! Your manuscript is now being handed over to our production team.

Kind regards,

on behalf of

Dr. Stanley Chinedu Eneh

Academic Editor

PLOS ONE